# Music intervention for sleep quality in critically ill and surgical patients: a meta-analysis

Ellaha Kakar ,[1] Esmée Venema,[2] Johannes Jeekel,[3] Markus Klimek ,[4] Mathieu van der Jagt[5]

► Prepublication history and supplemental material for this paper is available online. To view these files, please visit the journal online (http://dx.doi.org/10.1136/bmjopen-2020-042510).

EK and EV are joint first authors.

[1]Department of Surgery and Intensive Care Unit, Erasmus MC, Rotterdam, The Netherlands
[2]Maastricht University, Maastricht, The Netherlands
[3]Department of Surgery and Neuroscience, Erasmus MC, Rotterdam, The Netherlands
[4]Department of Anaesthesiology, Erasmus MC, Rotterdam, The Netherlands
[5]Department of Intensive Care Unit, Erasmus MC, Rotterdam, The Netherlands

**Correspondence to**
Ellaha Kakar;
e.kakar@erasmusmc.nl

## ABSTRACT

**Objective** Sleep disruption occurs frequently in hospitalised patients. Given the potential of music intervention as a non-pharmacological measure to improve sleep quality, we aimed to assess and quantify current literature on the effect of recorded music interventions on sleep quality and quantity in the adult critical care and surgical populations.

**Design** Systematic review and meta-analysis.

**Data sources** Embase, MEDLINE Ovid, Cochrane Central, Web of Science and Google Scholar.

**Eligibility criteria for studies** Randomised controlled trials assessing the effect of music on sleep quality in critically ill and surgical patients.

**Methods** The electronic databases were systematically searched from 1 January 1981 to 27 January 2020. Data were screened, extracted and appraised by two independent reviewers. Primary outcomes were sleep quality and quantity, assessed with validated tools. The Preferred Reporting Items for Systematic Reviews and Meta-Analyses guidelines were followed. Random effects meta-analysis was performed, and pooled standardised mean differences (SMDs) with 95% CIs were reported.

**Results** Five studies (259 patients) were included in qualitative (risk of bias) and quantitative analysis (meta-analysis). Pooled data showed a significant effect of recorded music on subjective sleep quality in the critical care and surgical population (SMD=1.21 (95% CI 0.50 to 1.91), p<0.01, excluding one non-English study; SMD=0.87 (95% CI 0.45 to 1.29), p<0.01). The SMD of 1.21 corresponded to a 27.1% (95% CI 11.2 to 42.8) increase in subjective sleep quality using validated questionnaires. A significant increase in subjective sleep quantity of 36 min was found in one study. Objective measurements of sleep assessed in one study using polysomnography showed significant increase in deeper sleep stage in the music group.

**Conclusions** Recorded music showed a significant improvement in subjective sleep quality in some critical care and surgical populations. Therefore, its use may be relevant to improve sleep, but given the moderate potential for bias, further research is needed.

**PROSPERO registration number** CRD42020167783.

### Strengths and limitations of this study

► This is the first comprehensive systematic review that studies the effect of music on sleep quality in critically ill and surgical patients.
► Patient-reported outcomes, which are of increasing interest and of clinical relevance, are assessed in qualitative and quantitative analyses.
► Only data from randomised controlled trials are included.
► The risk of bias is moderate to high and the heterogeneity is high, which makes it difficult to draw definite conclusions.

the Netherlands found that sleep quality and quantity were significantly affected in hospitalised patients compared with patients' habitual sleep at home.[5] Sleep disruption is prevalent in almost all surgical and critically ill patients.[5–8] Important preventable factors associated with sleep disruption are pain, noise, anxiety, stress, sedative and analgesic medication usage, immobility and severity of illness.[1 5 9–12]

In critically ill and surgical patients, poor sleep has been associated with postoperative delirium, cardiovascular events in high-risk patients, postoperative fatigue (leading to prolonged immobility), altered mental status,[13–16] prolonged mechanical ventilation, cognitive impairment, altered immune function and long-term psychological comorbidities.[17] Thus, improving sleep promotes good health by improving emotional (anxiety and stress)[18] and possibly physical recovery. This is especially important for intensive care unit (ICU) and surgical patients given the loss of tissue integrity.[19 20] Interventions in the hospital to improve sleep quality mainly include pharmacological interventions, such as benzodiazepines.[21–25] The Clinical Practice Guidelines for the Prevention and Management of Pain, Agitation/Sedation, Delirium, Immobility, and Sleep Disruption

## INTRODUCTION

Sleep disruption is common in hospitalised patients.[1–4] A large cross-sectional study in

in adult patients in the ICU strongly recommend avoiding these drugs, given their potential to induce delirium and substance dependence.[17 26 27] Therefore, non-pharmacological measures are advised to prevent sleep disruption in hospitalised patients.[17 28–36]

Recently, several studies have shown a significant positive effect of perioperative music interventions on anxiety, pain, sedative and analgesic medication requirement, and neurohormonal stress response.[37–39] Also, in the ICU population associations have been made between music interventions and decreased serum cortisol levels, positive effects on state anxiety, reduction in respiratory rate and systolic blood pressure, and decrease in sedative and analgesic requirement in mechanically ventilated patients.[40 41]

Given the potential of music intervention as a non-pharmacological measure to improve sleep quality, we aimed to assess and quantify current literature on the effect of perioperative recorded music interventions on sleep quality and quantity in the adult critical care and surgical populations.

## MATERIALS AND METHODS
### Protocol and registration
This systematic review and meta-analysis follows the Preferred Reporting Items for Systematic Reviews and Meta-Analyses guidelines (online supplemental file 1) and was registered in the PROSPERO database (https://www.crd.york.ac.uk/PROSPERO, as record number CRD42020167783).[42]

### Search strategy
A systematic search was performed together with a dedicated biomedical information specialist in the Embase, MEDLINE Ovid, Cochrane Central, Web of Science and Google Scholar databases using a standardised protocol,[43] including articles between 1 January 1981 (the year in which the first minimally invasive surgical intervention was performed, in order to avoid old literature which is not compatible with the current standard perioperative pain practice) and 27 January 2020. The search strategy included terminologies related to: sleep quality/insomnia, including sleep architecture (eg, rapid eye movement (REM) sleep) and music (eg, music therapy). The full search strategy per database is given in online supplemental file 2.

### Study screening and selection
Two reviewers (EK/EV) independently screened results of the search strategy on title/abstract to confirm adherence to the eligibility criteria. Studies were eligible when they investigated (P) critically ill and/or surgical patients aged 18 years or older receiving a (I) recorded music intervention (C) compared with a control group in order to assess the effect on (O) sleep quality and quantity in (S) a randomised controlled trial (RCT). PICOS is a mnemonic used in evidence-based medicine and stands for, respectively, patient, intervention, control, outcomes

and study type.[44] Critically ill populations included patients admitted to ICU or cardiac care unit (CCU), since CCU patients were also considered to have (potential) vital organ failure. Other eligibility criteria were: full text available and the study included in-hospital patients. Since, the above-mentioned studies included patients who have the following in common; in-hospital patients and a compromised physiology (eg, by performing an intervention, surgical procedure or having critical illness), which both have impact on the patients comfort and thus sleep, the data were suitable to be pooled. Studies involving live music performance were excluded, since live music performance consists of two interventions: the music and the interaction with the musician. Music intervention was defined as the use of recorded music consisting of melody, harmony and rhythm. Nature sounds were only included when used in addition to music. If studies compared music with multiple groups, the group without music most similar to the music group was chosen as control group (eg, if groups were 'scheduled rest' and 'standard care', 'scheduled rest' was chosen as control group if the music group also received a resting period). The data of the extra arms were therefore not used in this study. Full-text articles were discussed for admissibility. All disagreements between reviewers were resolved by discussion.

### Data collection process and items
Data were extracted and checked by the same two reviewers (EK/EV) independently according to a predesigned dataset. The following study characteristics were extracted: author, year of publication, study type, country of study, reason for admission (eg, surgery type), sample size, age (mean and SD), gender (% male), severity of disease of included patients, type and timing of sleep assessment, timing of the intervention, setting (eg, surgical ward, ICU and CCU), method of intervention delivery (eg, headphones and CD player), frequency and duration of the intervention, total duration of the intervention, music choice (eg, patient or researcher selected), type of control group and number of participants in the intervention and control group. Primary outcome measures were sleep quality and quantity measured after intervention or at the end of study period for both the control and intervention groups. If a study applied music intervention and collected data at multiple time points, only the final time point was used, since music could have a cumulative effect.

### Risk of bias assessment
Risk of bias within studies was assessed independently by two reviewers (EK/EV) using the Cochrane Risk of Bias tool for RCTs.[45] Incomplete data of ≥10% due to drop-out (attrition bias) was considered as high risk. If the study protocol was not available, the risk for selective reporting was considered as unclear. Publication bias was assessed by creating funnel plots, and the risk of bias across studies of the effect of music on sleep quality was assessed using

the Grading of Recommendations, Assessment, Development and Evaluations (GRADE) criteria.[45 46]

## Statistical analysis

Descriptive statistics were presented as means and their SD, counts (N) and percentages (%). For categorical age groups (eg, age 20–50 years=eight patients, 51–60 years=seven patients), an approximation of the mean was calculated by taking the central point of the range as the mean for each category and pooling these means for each trial separately, weighted to the sample size. The overall % of males in this review was calculated by averaging the % of males weighted to the sample size. If ranges were provided, these were used to calculate approximations for the SD by calculation one-fourth of the range of the data ((maximum-minimum)/4).[47] Studies were included for quantitative analysis if mean values and SDs of the sleep scores were reported. The primary analysis included meta-analysis for sleep quality and quantity. Standardised mean differences (SMDs) were calculated as summary statistics of the main outcome, and a random effects model was used to calculate the overall effect of music on sleep quality and quantity, accounting for between-study heterogeneity. The between-study variance was estimated using the restricted maximum likelihood method. The level of heterogeneity was assessed using the $I^2$ statistic. For clinical interpretation, the effect of the music intervention on sleep quality scores was expressed as a relative percentage increase, using a back-transformation of the acquired SMD that was described by the Cochrane Handbook for Systematic Reviews.[45] For this calculation, the SDs of the control groups were pooled to approximate the amount of variation in our patient population. Additionally, the SD of a study with a large sample assessing sleep was used to make a more accurate approximation of the amount of variation. Data were analysed using R V.3.6.3, and a two-sided p value of <0.05 was considered statistically significant. To calculate the increase in subjectively measured sleep quality based on the pooled SMD of the meta-analysis, a back-transformation was applied to the sleep quality as described by the Cochrane Handbook for Systematic Reviews.[45] For this back-transformation, the SDs of sleep were estimated by pooling the SDs of the control groups for the different assessment tools separately.

## Patient and public involvement

No patients involved.

## RESULTS

The search yielded 2127 articles leaving 1179 articles after removal of duplicates and 1146 articles after removal of articles before the year 1981. Title and abstract screening resulted in 10 articles for full-text assessment. Five studies were excluded for the following reasons: full text not available, use of guided imaging in combination with music, use of nature sounds and use of multiple interventions.

Subsequently, a total of five RCTs[48–52] were included in the qualitative and quantitative analysis (259 patients) (figure 1).

## Study characteristics

A detailed overview of patient and music intervention characteristics is presented in tables 1 and 2. Two studies were conducted at the ICU, two studies were conducted at the CCU and one study was performed in the surgical ward directly after ICU discharge. Apart from this study (Zimmerman et al[52]), no other surgical papers were found. The mean age of the study population was 62.4 years with a predominance of males (65.8%). Hansen et al[48] and Su et al[51] conducted the study at the ICU. The study of Su et al[51] included medical ICU patients, with an Acute Physiology and Chronic Health Evaluation II score of ≤25, and the study of Hansen et al[48] included surgical (59%) and medical (41%) ICU patients. Cheraghi et al[49] and Ryu et al[50] conducted their study at the CCU, respectively, including acute coronary syndrome (ACS) and percutaneous transluminal coronary angiography (PTCA) patients. The study of Zimmerman et al[52] was performed in the nursing ward after coronary artery bypass graft (CABG) surgery. Since the abstract of the paper of Cheraghi et al[49] was available in English, the study was included after title/abstract screening. The full text version was only available in the Persian language; the paper was translated by a statistician in the Erasmus MC with Persian as mother language. The study was included in this review due to acceptable methodology, which was not different than the other included studies.

Sleep quality was measured using the Richards-Campbell Sleep Questionnaire (RCSQ; 40%), the Verran and Snyder-Halpern Sleep Scale (VSH; 40%) and the Pittsburgh Sleep Quality Index (PSQI; 20%). Online supplemental file 3 gives an overview of the previously mentioned sleep questionnaire characteristics. Only in the study of Ryu et al sleep quantity was measured using patient questionnaires. In the study of Su et al,[51] sleep quality and quantity was also measured objectively, using the polysomnography (PSG). PSG is currently the golden standard in objectively measuring sleep variables, including total sleep time (TST), wake after sleep onset, sleep onset latency, REM latency, sleep efficiency, arousal index and percentage of TST spent in each sleep stage (N1, N2, N3 and REM).[53]

Music was mostly described as being soothing, sleep inducing or sedating (60%).[48 50 51] Music was selected by the researcher in four of the five studies (80%); only Zimmerman et al[52] provided the option of choice between five audiotapes according to the patients' preference (table 2). Music was mostly administered using headphones or earphones (60%). Duration of the intervention was on average 40 min per session and ranged from 30 to 53 min per day. Cheraghi et al[49] administered the music intervention during three consecutive evenings, just before bedtime, after admission to the CCU, and assessed sleep at baseline and every morning after an

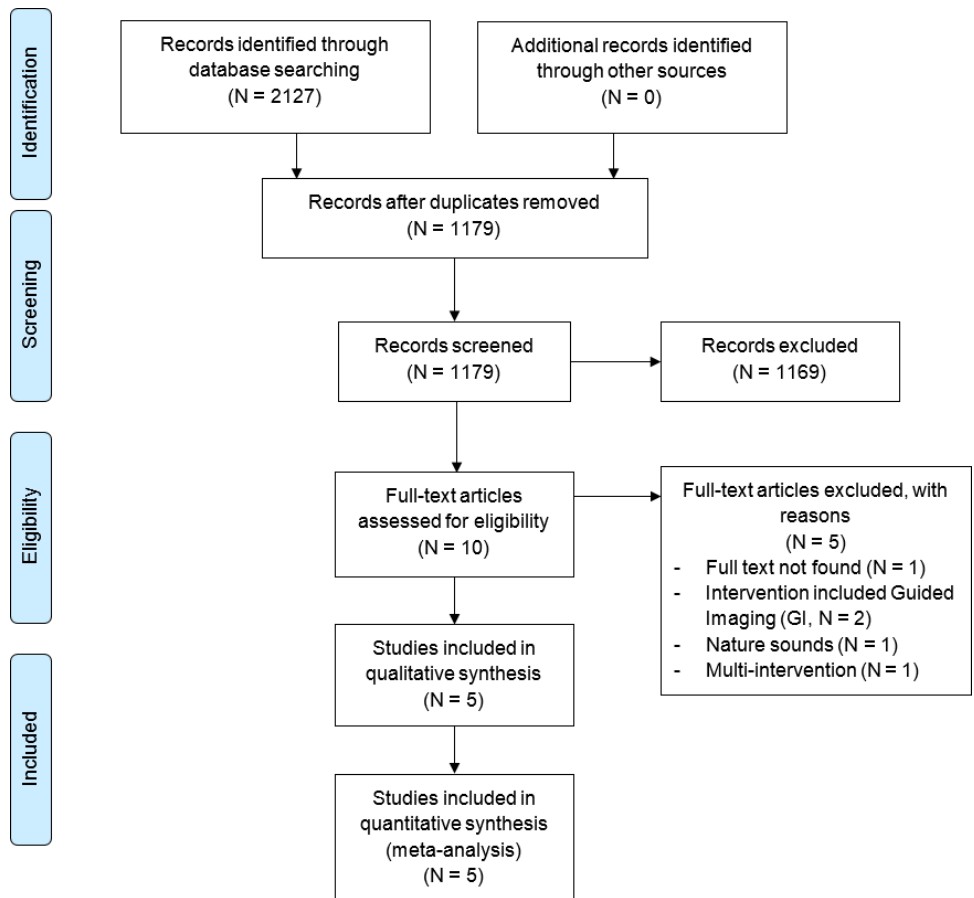

**Figure 1** PRISMA flor diagram. N indicates the number of articles. GI, Guided Imaging; PRISMA, Preferred Reporting Items for Systematic Reviews and Meta-Analyses.

evening of intervention. In the study of Su *et al*,[51] the intervention was only administered in the night of day 3 after admission to the ICU, and sleep was assessed at baseline (PSG and VSH) and PSG during the first 2 hours of sleep on the same night and VSH in the morning after the night of intervention. Hansen *et al* applied the intervention during the day on day 3 after admission and assessed sleep once immediately afterwards. Zimmerman *et al*[52] applied the intervention on the night of postoperative day (POD) 2 and assessed sleep at baseline and the morning of POD 3. Ryu *et al*[50] did not specify on which POD the music intervention and sleep assessment was performed. Baseline sleep assessment was carried out in only three studies;[49 51 52] no differences were found between the study groups in any of the studies based on baseline data. Control groups received either standard care (two studies), scheduled rest (two studies) or standard care with earplugs and eye shields (one study).

### Effect of music on subjectively measured sleep quality
Sleep quality scores of the final music session were pooled.[48–52] For the data of Cheraghi *et al*,[49] we applied reverse scoring since they used the PSQI, which was the only tool assessing sleep quality as better when scored lower on the scale. Pooling data resulted in an overall significant effect of recorded music on sleep quality in critical care and surgical patients (SMD=1.21 (95% CI 0.50 to 1.91), p<0.01). Figure 2A,B presents the forest and funnel plot. The study of Cheraghi *et al* showed a relative large effect compared with the other studies. Pooled data excluding the study of Cheraghi *et al* showed that the effect remained significant (SMD=0.87 (95% CI 0.45 to 1.29), p<0.01, figure 2C).

Since out meta-analysis eventually only included subjectively measures sleep assessment tools, the back-transformation was based on a recent Dutch study that assessed sleep in 194 patients using the RCSQ, which is a validated and reliable sleep quality assessment tool in the critically ill.[54] The reported median (IQR) for the RCSQ reported in this study was transformed to mean±SD using the methodology of Wan *et al*.[47] This resulted in an SD approximation of 2.24. Using the effect size of our meta-analysis, SMD=1.21, this resulted in a reduction of 2.71 (95% CI 1.12 to 4.28) on the RCSQ, which ranges from 0 to 10. This implies a 27.1% (95% CI 11.2 to 42.8) increase in sleep quality in the ICU due to a music intervention. In order to check if the effect in our study population approximates the effect calculated previously, we also applied the back-transformation using the pooled RCSQ SDs (n=51, SD=2.53) of the control groups in our study. This resulted in a reduction of 3.06 (95% CI 1.27 to 4.83) on the RCSQ, equal to 30.6% (95% CI 12.7 to 48.3) increase in sleep quality.

**Table 1** Study characteristics

| Study | Study type | Country | Reason for admission | Sample size | Mean age (SD) | | Gender (% male) | | Outcome assessment | Severity of disease of included patients |
|---|---|---|---|---|---|---|---|---|---|---|
| | | | | | Intervention | Control | Intervention | Control | | |
| Cheraghi et al[49] | RCT | Iran | ACS | 72 | 56.9 (10.5) | 62.1 (12.7) | 80.6 | 80.6 | Sleep quality (PSQI) | ▲ Oriented in person, time and place.<br>▲ Haemodynamically stable.<br>▲ No ventilation.<br>▲ History of hospitalisation; 55.6%. |
| Hansen et al[48] | RCT | Denmark | Surgical and medical ICU | 37 | 60.0 (18.0) | 65 (16.0) | 55.6 | 52.6 | Sleep quality (RCSQ) | ▲ GCS; ≥14.<br>▲ Communicable<br>▲ ICU LOS; 3 (3).<br>▲ Comorbidity; 59.0%.<br>▲ Planned hospitalisation; 49.0%. |
| Ryu et al[50] | RCT | South Korea | PTCA | 58 | 58.5 (15.0) | 59.9 (15.0) | 65.5 | 65.5 | Sleep quality (VSH), sleep quantity (questionnaire) | ▲ No ventilation.<br>▲ No use of sleep inducing or sedative medications. |
| Su et al[51] | RCT | Taiwan | Medical ICU | 28 | 62.4 (9.1) | 60.9 (10.8) | 57.1 | 64.3 | Sleep quality (VSH and PSG) | ▲ Communicable.<br>▲ Haemodynamically stable.<br>▲ ICU stay; ≥24 hours.<br>▲ Free from symptoms of current infection.<br>▲ APACHE II; 18.6 (3.8).<br>▲ On ventilation; 35.7%. |
| Zimmerman et al[52] | RCT | USA | CABG | 64 | 67.0 (9.9) | 67.0 (9.9) | 68.0 | 68.0 | Sleep quality (RCSQ) | ▲ Oriented in person, time, and place.<br>▲ Extubated within 24 hours.<br>▲ Hospital LOS 5.8 (2.3) days.<br>▲ Previous surgery; 90.0%. |
| Summary of all studies | | | | 259 | 60.8 (12.6) | 63.1 (11.2) | 68.0 | 64.4 | | |

ACS, acute coronary syndrome; APACHE, Acute Physiology And Chronic Health Evaluation; CABG, coronary artery bypass graft; GCS, Glasgow Coma Scale; ICU, intensive care unit; LOS, length of stay; PSG, polysomnography; PSQI, Pittsburgh Sleep Quality Index; PTCA, percutaneous transluminal coronary angioplasty; RCSQ, Richards-Campbell Sleep Questionnaire; RCT, randomised controlled trial; VSH, Verran and Snyder-Halpern Sleep Scale.

**Table 2** Music intervention characteristics

| Study | Location | N music group | Intervention | Intervention choice | Medium | Frequency (per day) × duration (min) | Total duration (min) | Timing intervention | Control | N control group | Timing sleep assessment |
|---|---|---|---|---|---|---|---|---|---|---|---|
| Cheraghi et al[49] | CCU | 36 | Non-vocal music, accepted by the society. | Researcher | Headphones | 3×45 | 135 | Every night after admission during three nights before sleep. | Standard care. | 36 | Baseline and every morning during three consecutive days. |
| Hansen et al[48] | ICU | 18 | MusiCure: soothing music, soft wind, bird twitter, ocean sound and music instruments. | Researcher | Loudspeaker in the ceiling of patients' bed | 1×30 | 30 | During the day on day 3 after admission. | Standard care during rest. | 19 | Once directly after intervention. |
| Ryu et al[50] | CCU | 29 | Sleep-inducing music: nature sounds, delta wave control music and Goldberg variations BWV+eye bandages. | Researcher | Earphones | 1×53 | 53 | Postoperative in the evening, timing NR. | No music with earplugs and eye shield. | 29 | Once in the morning after intervention. |
| Su et al[51] | ICU | 14 | Sedating piano music: smooth rhythm to achieve a relaxing effect, a tempo of 60–80 beats/min, minor tonalities, smooth melody lines and no dramatic changes in volume and rhythm. | Researcher | CD player | 1×45 | 45 | Night of day 3 after admission | Same procedure, no music. | 14 | Baseline VSH and PSG, PSG first 2 hours of night of intervention and VSH once in the morning after intervention. |
| Zimmerman et al[52] | Ward | 32 | Country western instrumental, fresh aire (Mannheim Steamroller), winter into spring (George Winston), prelude and comfort zone (Steven Halpern) (facilitates relaxation). | Option out of five audiotapes, in case of no preference, Halpern was chosen | Headphones | 2×30 | 60 | Afternoon or early evening on POD 2–3. | Scheduled rest. | 32 | Baseline and once in the morning on POD 3. |

BWV, Bach-Werke-Verzeichnis; CCU, coronary care unit; ICU, intensive care unit; NR, not reported; POD, postoperative day; PSG, polysomnography; VSH, Verran and Snyder-Halpern sleep scale.

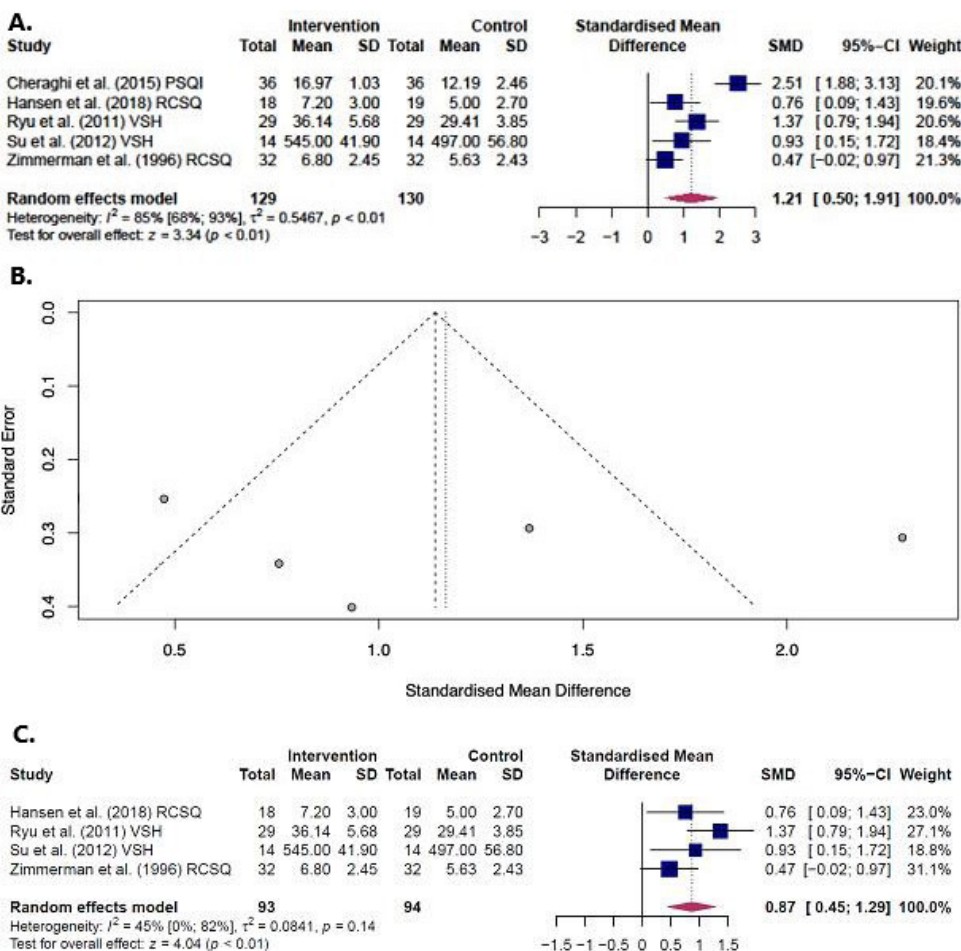

**Figure 2** (A) Forest plot assessing the effect of music on subjective sleep quality. (B) Funnel plot assessing the effect of music on subjective sleep quality. (C) Forest plot assessing the effect of music on subjective sleep, excluding the paper of Cheraghi *et al.* SMD, standardised mean difference.

### Effect of music on subjectively measured sleep quantity

Meta-analysis on the effect of music on sleep quantity could not be performed since it was only assessed by the study of Ryu *et al*[50] in PTCA patients. This study found a statistically significant higher sleep duration in the music group (n=29) compared with the control group (n=29), 279.31±43.99 and 243.10±42.68, respectively, with a p value of 0.002.

### Effect of music on objectively measured sleep

As mentioned before, only the study of Su *et al* measured sleep objectively using the PSG. In this study, they found a significant difference in the N2 (Wald $\chi^2$=6.03, p=0.014) and N3 (Wald $\chi^2$=7.02, p=0.008) sleep stages, indicating that the music group had a shorter N2 sleep stage and a longer N3 sleep stage.

### Risk of bias assessment

The overall risk of bias was moderate to high (figure 3). All included studies reported the use of randomisation, but a high risk of selection bias was considered due to insufficient details regarding random sequence generation in two studies[48 52] and allocation concealment in four studies.[49–52] Due to the type of intervention and the

subjective outcome assessment blinding of participants and personnel was not possible, therefore performance and detection bias were considered high in all studies. Attrition bias was considered low in all studies since none of the studies had a drop-out rate of ≥10%. A full study protocol was missing (assessed by checking reported registration number of the trial in the paper and registers for clinical trials (eg, ClinicalTrials.gov) for all included studies leading to an unclear risk for reporting bias. Two included studies (20%) had a high risk of bias due to other reasons; Cheraghi *et al*[49] excluded patients if listening to the music induced anxiety or brought up bad memories, which could have contributed to an overestimation of the effect of music intervention on sleep quality, and Hansen *et al*[48] measured sleep quality during daytime rest with the RCSQ, which is only a validated measurement tool for sleep quality after nocturnal sleep. Figure 4 presents the summary of the risk of bias assessment.

### Grade certainty rating

Risk of bias was moderate to high for sleep quality. Imprecision was considered medium since the effect size in the 95% CI of the pooled estimate ranged from medium to

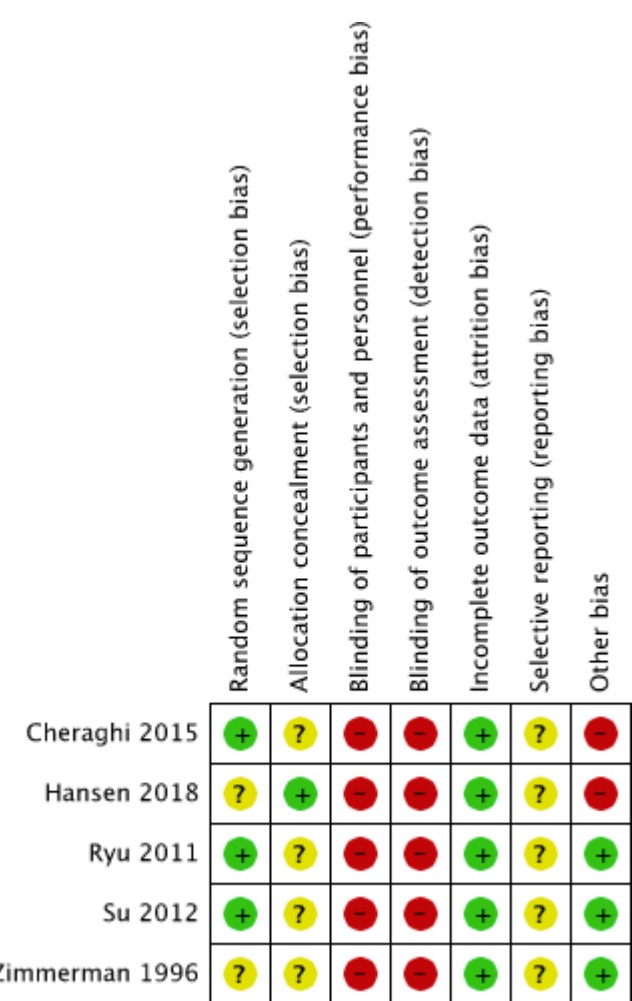

**Figure 3** Risk of bias assessment.

and was non-significant ($I^2$=45% (95% CI 0% to 82%), $T^2$=0.08, p=0.14). Meta-regression, to assess the reason for high heterogeneity, could not be performed since there was a limited number of papers available. Directness was considered high as studies directly investigated music in the population of interest, the intervention was mostly applied on the third day after ICU/CCU admission or after surgery and the reported outcomes were critical for decision making (eg, if sleep quality was low, administration of sleep medication could be considered). Scattering in the funnel plot is symmetrical; therefore, the risk for publication bias is considered low. In conclusion, we rate the GRADE certainty rating as moderate.

## DISCUSSION

This meta-analysis showed a significant beneficial effect of recorded music intervention on subjectively measures of sleep quality in critically ill and post-CABG patients admitted to the ward (SMD=1.21 (95% CI 0.50 to 1.91), p<0.01), when on average 40 min (range 30–53) per session/day of music is applied. The overall risk of bias was moderate to high.

These findings are in line with the current literature.[29 55–57] Hu et al[29] reviewed literature for the effect of non-pharmacological interventions on sleep quality in the ICU. Although they could not pool data and their quality of evidence was low, they found that music intervention might positively influence sleep quality. Also, Feng et al[55] studied the effect of music and other non-pharmacological interventions (eg, music combined with other interventions and acupuncture) on primary insomnia (insomnia after ruling out several other conditions such as psychiatric (depression and anxiety), medical (pain), circadian (phase-delay syndrome) or other sleep disorders). They found that solely music intervention had the highest ranking and seemed to offer clear advantages on sleep quality. Our findings are also in line with the current literature on the effect of music on other outcomes in the critically ill population. These studies show effectiveness of music on anxiety, pain, vital parameters, inflammatory markers and medication requirement.[40 41 58–60]

high, which could influence clinical decision making. Consistency was considered moderate to high since individual studies included in the meta-analysis showed a positive influence of music on sleep quality, the 95% CIs overlapped, but the heterogeneity was high. Heterogeneity was moderate to high and statistically significant for sleep quality ($I^2$=85% (95% CI 68% to 93%), $T^2$=0.5, p<0.01), possibly caused by the differences in patient characteristics. After the exclusion of the outlier study of Cheraghi et al from the meta-analysis, heterogeneity decreased

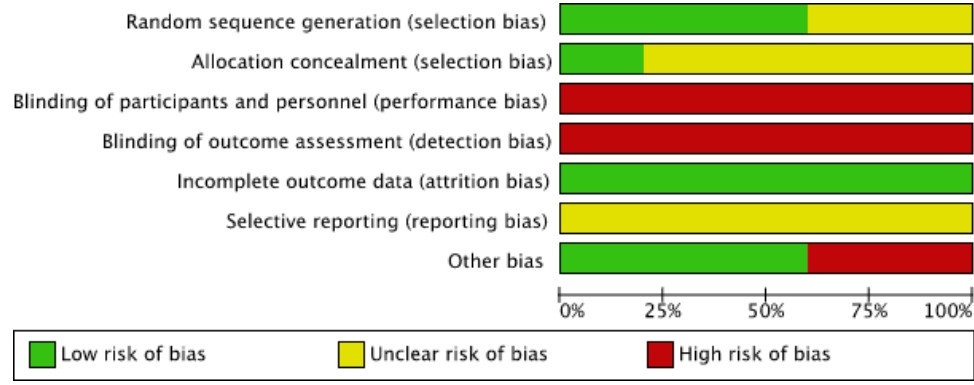

**Figure 4** Summary of risk of bias.

Sleep architecture in critical care patients is characterised by decreased TST, sleep divided into short discrete episodes, sleep occurring mostly during daytime, increased amount of light sleep stages and decreased amount of deeper sleep stage.[8 17 61–64] This leads to loss of the restorative and resting function of sleep, which is achieved by deeper sleep stages.[40 61 63 65] Sleep disturbances have been associated with cognitive impairment, including delirium, prolonged duration of mechanical ventilation, altered immune function and long-term psychological comorbidities.[8 17 66–69] Furthermore, acutely developed sleep disturbances in the ICU can persist after discharge and is the most common stressful factor experienced by patients.[61 70–73] Furthermore, the administration of sedative and analgesic medication, aimed to improve patient comfort, may negatively impact sleep quality.[5 10–12 74–78] Therefore, preventing sleep disturbances in critically ill patients is highly important. The one study in our review that assessed sleep objectively found that music significantly increased the deeper sleep stage N3 and decreased the light sleep stage N2, inevitably strengthening the results of our meta-analysis.[51]

We made an approximation of 27.1% increase in sleep quality if music intervention is applied based on our pooled data. Several studies comparing benzodiazepines to a placebo showed *even* lower efficacy, ranging from 12.9% to 21.4% improvement.[79–82] This indicates that music may have a similar or even better effect on sleep quality compared with drug therapy with benzodiazepines, while the patient is not exposed to their harmful side effects and risk of dependency. Therefore, the use of recorded music in hospitalised patients can be considered a clinically relevant intervention. Only one paper in this study assessed effect of music on sleep quantity and found a significant increase in sleep duration of 36 min in a sample of 58 patients.[50] Krenk *et al*[81] found that zolpidem shortened TST with 28 min compared with the placebo group; this finding was not significant. Simons *et al*[82] found a significant extension of 57 min when using temazepam in healthy volunteers. Also, Sharma *et al* measured an increased TST when zolpidem was used; unfortunately, effects on deeper sleep stages were not found. Of note, extension of the TST does not necessarily lead to better sleep, since better sleep depends more on the proportion of restorative sleep.

The physiology behind the effect of music on sleep remains unclear. Several theories have been proposed on the effect of music on sleep including: rhythmic entrainment, masking and distraction.[57] It is important to recognise that the effect of music also entails the neurophysiological and psychophysiological levels.[83–86] Literature indicates that music induces the anxiolytic effect through suppression of the sympathetic nervous system.[87 88] Also, music stimulates the release of endorphins by activating memory and the limbic system, which plays an important role in emotional well-being.[58 89] More specifically, the nucleus accumbens is activated leading to increased dopamine release and deactivation of areas in the brain related to stress and cortisol signalling.[90] Furthermore, several studies have suggested positive effects of vagus nerve stimulation[91] on objective and subjective sleep parameters in patients with epilepsy.[92 93] Unfortunately, in spite of these theoretical considerations, we did not find literature on the mechanistic effect of music on sleep architecture. Thus, sleep disturbances are multifactorial, and non-pharmacological interventions should focus on tackling multiple factors (anxiety, pain, stress, etc) affecting sleep at the same time.

The GRADE certainty was rated moderate, which supports the estimated clinical relevant effects. Unfortunately, the relatively small number of studies and sample sizes lead to uncertain reliability and validity of this study, which is also seen in the current literature regarding the effect of music on sleep in other populations.[79–81]

## Strengths and limitations

The sleep assessment tools used in the included studies of this meta-analysis were all validated, reliable and easily applicable. All tools used in the studies included in the meta-analysis are validated and reliable self-reporting questionnaires assessing sleep health of the previous night and are used to assess sleep quality in the studies included in this meta-analysis. The tools are widely used to measure sleep quality in hospitalised patients and thus are deemed conceptually to perform meta-analysis.[94] An important limitation of this review is the limited amount of studies included for quantitative analysis. Second, heterogeneity was high. Third, there was a moderate to high risk of bias due to insufficient information on random sequence generation and allocation concealment, the primary outcome including subjective patient-reported questionnaires, and due to blinding of participants in a music intervention study not being feasible. Although, Chlan *et al*[95] tried to avoid bias due to non-blinding in a large clinical trial on music intervention, that found a decrease in anxiety in the music group, by including an extra control group wearing noise-cancelling headphones (without music).

## Future research recommendations

We suggest future research should consist of high-quality RCTs with the use of objective tools for sleep assessment, as recorded music seems effective and clinically relevant, in order to make more definite conclusions regarding the effect of recorded music on sleep. Since until now relatively small studies with a high variability in the music 'dose' are conducted, we recommend future studies to focus on larger sample sizes with a high methodological quality in order to avoid a substantial risk of bias. Studies with music interventions should report the type of music, timing, duration and frequency of the intervention and sleep assessment in their studies with a validated and reproducible tool. Our analyses suggest that a minimum of 30 min per day/session is sufficient in order for the music intervention to be effective for the sleep quality. This minimum of 30 min of music per session/day is

also consistent with the current literature.[37 40 59] Previous studies can serve as a guide for future studies.[96]

## Conclusion

This systematic review and meta-analysis of randomised controlled trials showed that recorded music interventions significantly increase sleep quality in the critically ill, ACS or PTCA patients and after cardiac surgery for coronary artery disease. Music is easily applicable and has no risks and side effects and should therefore be considered as a suitable non-medicinal alternative for sleep quality improvement in these patient groups. Since the clinical trials performed until now are small and of low quality, we suggest larger and high-quality randomised clinical trials for future research, including broader patient populations.

**Acknowledgements** The authors would like to thank W Bramer, biomedical information specialist of the Medical Library at the Erasmus University Medical Centre, for his assistance in the literature search and J van Rosmalen, assistant professor at the Department of Biostatistics of the Erasmus University Medical Centre, for his assistance in the statistical analysis.

**Contributors** JJ conceived the study idea. EK and EV coordinated the systematic review. EK and EV screened abstracts and full texts. EK and EV wrote the first draft of the manuscript and judged risk of bias in the studies. EK, EV, MvdJ, MK and JJ interpreted the data. EK, EV, MvdJ, MK and JJ critically revised the manuscript. EK, EV, MvdJ, MK and JJ had full access to all of the data in the study and can take responsibility for the integrity of the data and the accuracy of the data analysis.

**Funding** The authors have not declared a specific grant for this research from any funding agency in the public, commercial or not-for-profit sectors.

**Competing interests** None declared.

**Patient consent for publication** Not required.

**Provenance and peer review** Not commissioned; externally peer reviewed.

**Data availability statement** Data sharing not applicable as no datasets generated and/or analysed for this study.

**ORCID iDs**
Ellaha Kakar http://orcid.org/0000-0002-7472-308X
Markus Klimek http://orcid.org/0000-0002-0122-9929

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
