## [Reviewer comments · BMJ Open]

ARTICLE DETAILS

TITLE (PROVISIONAL)	Music intervention for sleep quality in critically ill and surgical patients: a meta-analysis
AUTHORS	Kakar, Ellaha; Venema, Esmée; Jeekel, Johannes; Klimek, M; van der Jagt, Mathieu

VERSION 1 – REVIEW

REVIEWER	michele umbrello AO San Paolo - Polo Universitario
REVIEW RETURNED	15-Sep-2020

GENERAL COMMENTS	In this systematic review, Kakar et al investigate the effect of music intervention on sleep quality in a population of critically ill and surgical patients. The topic is interesting, important in its focus, and the authors investigated patient-centered outcomes such as the quality of sleep using validated tools. The authors found how music interventions significantly improved sleep quality. The meta-analysis seems well-conducted, the aim of the paper is well stated and the methods are adequately described. The introduction shortly introduces the topic. I have only some minor comments: - other systematic reviews and meta-analysis have been conducted in mechanically ventilated and critically ill patients, which investigated different outcomes associated with music interventions. Can the authors cite those researches and discuss the difference of their study in more detail?- the patients enrolled in the current study seem to be of low severity (CABG patients, post PTCA..). Can the authors provide more details as for the case-mix enrolled, such as severity indices (SOFA, SAPS II or APACHE), duration of mechanical ventilation or ICU stay and mortality? the relevance of the results and their generalizability should be interpreted in the light of such data- the authors nicely describe the architecture of sleep and the effect of critical illness on sleep architecture. Are they aware of any mechanistic study on the effect of music interventions on sleep architecture? Please discuss this issue
---

REVIEWER	Michele Balas Ohio State University College of Nursing, USA
REVIEW RETURNED	01-Oct-2020

GENERAL COMMENTS	Thank you for giving me the opportunity to review this important and timely study. The authors conducted a systematic review and meta-analysis exploring the effects of music intervention on sleep quality and quantity in the adult critical care and surgical populations. A total of 5 RCTs were included in analysis of the effect of music on sleep and 1 RCT evaluated the effect of music on sleep quantity. Overall, the paper is well written. It addresses a non-pharmacologic intervention and topic that is of great interest to practicing ICU clinicians. While the number of included trials (5) and participants (259) are indeed quite small, the reported outcomes appear consistent and the manuscript contributes to the growing body of knowledge in this area. I only have a few minor concerns/questions for the authors: 1) The number of databases searched seems low for a comprehensive systematic review. I am curious to know why CINAHL was not searched ? I would assume that this would be an area of interest to nurses. Is it possible that this data is captured in the search strategy in some way? 2) I am not familiar with the process used for the clinical interpretation (i.e., using data from the recent Dutch study). Is this normal/customary practice to do in a meta analysis? Why did the authors decide to do this? 3) The discussion section could be shortened/revised. While the authors make some excellent points in this particular section, I would suggest that the authors consider revising the first paragraph to summarize the overall study findings (effects, quality of data) before discussing how the results are consistent with current literature. I also believe that it may be worth discussing the "dose" of the intervention here. Meaning, the studies seem to vary widely in the frequency and total duration of music session.
--

REVIEWER	Irina Chis Ster St George's University of London
REVIEW RETURNED	25-Nov-2020

GENERAL COMMENTS	This is a well written communication although I have doubts about the conclusions of the paper. I can see a massive variability in the means of the outcomes in the table and, whilst I am aware that the analysis uses their standardized forms, can the authors clarify the rather very large scale of those measurements? The Iran study seems to stand out - can the authors detail explain why it seems to be far away from others? I am not sure about the pooled estimate given the amount of variability in the data. The analyses follows all the necessary steps but the whole point of conducting a meta -analysis is to identify and explain all the potential sources of variability rather than making sense of the pooled estimate.
--

REVIEWER	RUTH PICKERING UNIVERSITY OF SOUTHAMPTON retired
REVIEW RETURNED	08-Dec-2020

GENERAL COMMENTS	This paper reports a meta analysis of music therapy in critically ill
---

	and surgical patients which was based on 5 RCTS. I have the following specific points to raise. 1 Abstract, Objectives, description of intervention as perioperative. The timing of the intervention was not clear, starting from this point – not all the patients included in the constituent trials had operations, and the eligibility criteria didn't specify that patients had to be surgical. The descriptor perioperative is used in the statement of objectives at the end of the Introduction as well. Reading the Introduction, I was thinking that the intervention would be happening in the evening or maybe even while participants were asleep. This point wasn't clear in table 2. I think it was described in the text page 6, lines 10-30, for most of the trials. The information would be easier to assimilate if the descriptions in the second column of table 2 were expanded to include time of day, length of therapy on each occasion, number of occasions, and timing in relation to operation where appropriate. The issue of timing should be touched upon in the Introduction – that is whether its at night/evening or could be during the day– and it should be clear that the literature search put no restrictions on this. I think its only in the Discussion that they describe what constitutes poor sleep – though its mentioned in the Introduction there is no description. I was thinking about people sleeping during the day, presumably the music therapy should be given around bedtime to reduce daytime sleeping. 2 Abstract, line 32/33, qualitative and quantitative analysis. In the context of a systematic review this might be taken to refer to narrative review and quantitative meta-analysis respectively. Whereas I think they are referring to analysis of the quality and quantity of sleep. This should be rephrased. 3 There was also a lack of clarity about the patient group the review is meant to relate to. On page 6, lines 24- 26, they mention restricting the literature search to 1981 onwards, the year when minimally invasive surgery was introduced. They don't mention anywhere that surgical patients should have had minimally invasive surgery, and it doesn't apply to non-surgical patients. Does this mean that music therapy isn't appropriate for patients having more invasive surgery? or is it that it would be difficult to carry out a trial in this wider group? 4 Page 4, search strategy section. They should broadly list the various items in the strategy in the main text, in addition to giving the source specific code as supplementary material. In particular they should detail the strategy in relation to language. Looking at the supplementary material it appears the search was limited to English language papers. But on page 9, lines 41-45, they mention that Cheraghi et al was written in Persian (but one of the authors could read it). In which case the search wasn't limited to English language papers. This is especially an issue as from Figure 1 Cheraghi et al appears to be one of the worst with respect to bias and from Figure 2 it stands out from the others with respect to the effect size reported and may be responsible for the heterogeneity found. Had this trial been excluded the overall SMD would have fallen from 1.21 to 0.87. 5 Page 6, lines 41-48. Please could you explain what PICOS stands for. 6 Page 6, line 53/4. Studies involving live music were excluded. Please could you give a justification as to why such trials don't match the objectives of the current study. 7 Page 7, line 22-24. Abstracting information on population size. From Table 1, heading to column 5, its clear that by population size they mean the size of the trial. Usually population refers to a wider group from which people are selected to participate in the study. So it would be clearer on page 7 (and in table 2) to refer to the size of
--	---

the trial. When they describe the eligibility criteria its possible to include a trial with more than two groups – possibly a third arm could have a different intervention to improve sleeping – and in such a case only the arms closest to music therapy and control would be included in the meta analysis. I notice from Tables 1 and 2 that the “population size” is equal to the total of participants in the music therapy and control arms in all cases. Does that mean that none of these trials had additional arms that weren’t included, or does it mean that “population size” means the total of the music therapy and control arms, and not the size of the trial (in groups with more than two arms)?

8 Page 7, lines 37-41, the selection of data relating to the final time point to include in the study, because music therapy could have a cumulative effect. The number of participants still in trial follow-up tends to decrease with successive follow-ups. They do say somewhere that attrition in the trials was low (less than 10%), but perhaps they would have had analysis based on a larger number if they had selected an earlier timepoint. Another point is that though the impact of therapy might be expected to be cumulative with therapy sessions, it might also be expected to tail off after therapy has finished, and it might have been preferable to have planned to take the follow-up closest to the end of therapy. The timing of the assessment used, is described on page 10, lines 10-30 (the section that also described the timing of therapy). It would be easier to understand the timing of the assessment used if this was included in Table 1 – perhaps in the outcome assessment column. The way the timing of assessment was described on lines 10-30, didn’t sound as though there were multiple timepoint and the final one was selected as stated on page 7, lines 37-41.

9 Page 8, lines 14-18, how mean age was obtained where age was given in categories. They say the mean for each category was calculated and pooled. But if only numbers in each category are available category means can’t be calculated. They could have taken the central point of the age range in each category and averaged these, weighting by category frequencies? I assumed this related to the summary ages and genders at the bottom of Table 1. Did they obtained the trial specific mean ages then average over the 5 trials – and was this averaging weighted according to the size of the trial? They could also mention how the overall % male were obtained – this could also have been obtained by averaging trial specific %s.

10 Page 8, line 22, quantitative analysis. Its not clear whether they mean the analysis of the quantity of sleep here, or whether quantitative analysis is being used to mean meta analysis itself (as in my point 2).

11 Page 8, line 28/29, the choice to perform random effects meta analysis. This is most sensible when there are a large number of studies to be combined. Here they have only 5 (4 if Chereghi et al is excluded). A small number of studies is often taken as a reason to do fixed effects meta analysis. Did a fixed effects analysis result in different conclusions?

12 Page 8, line 24/5 onwards the main method of meta analysis. I would imagine that many of the constituent trials reported baseline values of the instruments chosen as their outcome measure, taken at the point of randomisation before the start of intervention. Was anything done with such information? IF not they should state that this information was not used.

13 Page 8, line 30/31. The “in” as in “the in between study variance was estimated...” can be omitted. The between study variance won’t be well estimated with only a few studies.

	14 Page 8, lines 37-47, the description of clinical interpretation. I didn't understand exactly what was done here. The description of the method continued on page 11, lines 4-7 in the Results section. I thought a standardised mean difference could be interpreted easily as the size of the difference in terms of SD between the groups. The methods described lead to their estimate of a 27.1% increase in sleep quality, the focus of conclusions (eg page 14, line 9/10) and in the Abstract line 39 where it is stated as a 26% increase. I felt the methods described were quite involved and based on assumptions – couldn't they concentrate on the SMID finding? 15 Page 9, lines 50-56, the description of the instruments used to measure outcome in the trials. It would be helpful if some description of these tools were given. In the Discussion there is a description of the various aspects of poor sleep quality. I wanted to know whether these instruments (and the patient questionnaire used by Ryu et al) measured the same underlying constructs. In particular I note from Table 1 that Chereghi et al was the only one using the PSQI, if this assesses different aspects to the others it might explain the extreme result from this trial. If the instruments don't measure the same thing they shouldn't be combined, the authors should address whether they are similar enough to combine. 16 Page 10, line 22. I don't think the acronym POD has been spelt out in the text. I notice several acronyms are spelt out in footnotes to the tables, but I think they should be spelt out in the text as well. 17 Page 10, lines 45-50. The analysis excluding the Chereghi trial. It would be useful at this point to comment on the heterogeneity results (i^2 and t^2). In particular they could report corresponding results from the analysis excluding Chereghi et al. Was there an indication of heterogeneity after the trial was removed? Given that this trial did not meet the stated eligibility criteria (it was not published in English), it might be better to emphasise the results excluding this trial in the Abstract and conclusions. 18 Page 11, lines 5-10, Chereghi et al excluding participants where the therapy induced anxiety etc. Could they state the numbers of participants that were excluded for this reason. IN connection to my point 7 above relating to the population size variable, the size of the trial. Do the numbers given in Tables 1 and 2 relate to the number of participants randomised in the constituent trials, or the numbers included in this meta analysis? It would be possible to include both in the tables. 19 Table 1. Please could they include the reference number for each trial after the study names in column 1.
--	--

VERSION 1 – AUTHOR RESPONSE

Reviewer: 1

Dr. Michele Umbrello, San Paolo University Hospital

1. other systematic reviews and meta-analysis have been conducted in mechanically ventilated and critically ill patients, which investigated different outcomes associated with music interventions. Can the authors cite those researches and discuss the difference of their study in more detail?

Response: *We would like to thank the reviewer for this comment. We have added a section in the second paragraph of the Discussion including these references:*

“Our findings are also in line with the current literature on the effect of music on other outcomes in the critically ill population. These studies show effectiveness of music on anxiety, pain, vital parameters, inflammatory markers, and medication requirement.”

2. the patients enrolled in the current study seem to be of low severity (CABG patients, post PTCA..). Can the authors provide more details as for the case-mix enrolled, such as severity indices (SOFA, SAPS II or APACHE), duration of mechanical ventilation or ICU stay and mortality? the relevance of the results and their generalizability should be interpreted in the light of such data

Response: *Thank you for this important comment. We agree that severity of disease of patients included varies between studies and that this should be clarified in the manuscript. We have added the severity indices, when reported, as a separate column named “Severity of disease of included patients” in Table 1. Unfortunately, no data were reported on duration of mechanical ventilation and mortality.*

3. the authors nicely describe the architecture of sleep and the effect of critical illness on sleep architecture. Are they aware of any mechanistic study on the effect of music interventions on sleep architecture? Please discuss this issue

Response: *Unfortunately, there are no studies, that we are aware of, on a convincing mechanistic effect of music on sleep architecture. We added the marked sentence in the Discussion to specify this:*

“More specifically, the nucleus accumbens is activated leading to increased dopamine release and deactivation of areas in the brain related to stress and cortisol signaling. Furthermore, several studies have suggested positive effects of vagus nerve stimulation (VNS) on objective and subjective sleep parameters in epilepsy patients. Unfortunately, in spite of these theoretical considerations, we did not find literature on the mechanistic effect of music on sleep architecture.”

Reviewer: 2

Dr. Michele Balas, Ohio State University College of Nursing

1. The number of databases searched seems low for a comprehensive systematic review. I am curious to know why CINAHL was not searched? I would assume that this would be an area of interest to nurses. Is it possible that this data is captured in the search strategy in some way?

Response: *We would like to thank the reviewer for the comment. The librarian involved who developed the search strategy has done extensive research on optimized search strategies in databases for Systematic Reviews (<https://pubmed.ncbi.nlm.nih.gov/29208034/>). The conclusion of his research was that the most optimal combination is Embase, Medline, Web of science and Google Scholar. Therefore, these are the databases that have been used in this SR. The librarian has investigated the added value of CINAHL, but he found that it only added relevant references that had not been retrieved by the other databases in reviews, where the topic was not just of interest to nurses, but especially was about nurses (such as nurses role, or compassion fatigue among nurses). Therefore we did not include CINAHL in the search.*

2. I am not familiar with the process used for the clinical interpretation (i.e., using data from the recent Dutch study). Is this normal/customary practice to do in a meta analysis? Why did the authors decide to do this?

Response: *Chapter 12.6.4 of the Cochrane handbook for systematic reviews describes the back-transformation as follow: "12.6.4 Re-expressing SMDs using a familiar instrument. The final possibility for interpreting the SMD is to express it in the units of one or more of the specific measurement instruments. Multiplying a SMD by a typical among-person standard deviation for a particular scale yields an estimate of the difference in mean outcome scores (experimental versus control) on that scale. The standard deviation could be obtained as the pooled standard deviation of baseline scores in one of the studies. To better reflect among-person variation in practice, it may be preferable to use a standard deviation from a representative observational study. The pooled effect is thus re-expressed in the original units of that particular instrument and the clinical relevance and impact of the intervention effect can be interpreted. However, authors should be aware that such back-transformation of effect sizes can be misleading if it is applied to individual studies rather than for a summary measure of effect (Scholten 1999). Consider two studies that did use the same instrument and observed the same effect, but observed different among-participant variability (perhaps due to different inclusion criteria). Then back-transformations using the different standard*

deviations from these studies would yield different sizes of effect for the same scale and the same effect.”

We added the following to the statistical analysis section in the Methods:

“To calculate the increase in subjectively measured sleep quality based on the pooled SMD of the meta-analysis, a back-transformation was applied to the sleep quality as described by the Cochrane Handbook for Systematic Reviews.⁴⁵ For this back-transformation, the SDs of sleep were estimated by pooling the SDs of the control groups for the different assessment tools separately.”

3. The discussion section could be shortened/revised. While the authors make some excellent points in this particular section, I would suggest that the authors consider revising the first paragraph to summarize the overall study findings (effects, quality of data) before discussing how the results are consistent with current literature. I also believe that it may be worth discussing the "dose" of the intervention here. Meaning, the studies seem to vary widely in the frequency and total duration of music session.

Response: *We shortened the Discussion and we added a short summary of the findings, music dose, and quality of the evidence in the first paragraph of the discussion as follows:*

“This meta-analysis showed a significant beneficial effect of recorded music intervention on subjectively measures of sleep quality in critically ill and post-CABG patients admitted to the ward (SMD= 1.21 [95% CI 0.50; 1.91], $p < 0.01$), when on average 40 minutes (range 30 – 53) per session/day of music is applied. The overall risk of bias was moderate to high.”

Reviewer: 3

Dr. Irina Chis Ster, St George's University of London

Comments to the Author

1. This is a well written communication although I have doubts about the conclusions of the paper. I can see a massive variability in the means of the outcomes in the table and, whilst I am aware that the analysis uses their standardized forms, can the authors clarify the rather very large scale of those measurements?

Response: *Thank you for taking the time in revising this manuscript.*

The Cochrane handbook of systematic reviews states the following:

“10.5.1 Which effect measure for continuous outcomes?”

The two summary statistics commonly used for meta-analysis of continuous data are the mean difference (MD) and the standardized mean difference (SMD). Other options are available, such as the ratio of means (see Chapter 6, Section 6.5.1). Selection of summary statistics for continuous data is principally determined by whether studies all report the outcome using the same scale (when the mean difference can be used) or using different scales (when the standardized mean difference is usually used). The ratio of means can be used in either situation, but is appropriate only when outcome measurements are strictly greater than zero. Further considerations in deciding on an effect measure that will facilitate interpretation of the findings appears in Chapter 15, Section 15.5.”

This method is especially designed so that we are able to provide level-1 evidence (meta-analysis) for studies that actually measure the same (subjective sleep quality) but do so in different manners (all the different sleep questionnaires).

2. The Iran study seems to stand out - can the authors detail explain why it seems to be far away from others?

Response: *This paper is the only paper that used the PSQI as a sleep assessment method, which may partly explain this outlier. To test the validity of our findings, we performed a leave-one-out analysis as described in the manuscript in the results. Leaving out this study does not alter the conclusions. We would like to refer to the Results (Effect of music on subjectively measured sleep quality) of the main text for the leave-one-out analysis.*

3. I am not sure about the pooled estimate given the amount of variability in the data. The analyses follows all the necessary steps but the whole point of conducting a meta -analysis is to identify and explain all the potential sources of variability rather than making sense of the pooled estimate.

Response: *We agree with the reviewer, that there is a high variability in the data, and we address this heterogeneity at several points in the manuscript, especially when judging possible bias and applying the GRADE-criteria. However, we also see that the effects of music on sleep quality are consistent since the SMD if all of the studies included in the meta-analysis is positive, indicating and*

effect of the intervention. We would like to refer to the response on the previous two comments of this reviewer regarding the variability.

Reviewer: 4

Dr. Ruth Pickering, University of Southampton

Comments to the Author

This paper reports a meta analysis of music therapy in critically ill and surgical patients which was based on 5 RCTS. I have the following specific points to raise.

1a Abstract, Objectives, description of intervention as perioperative. The timing of the intervention was not clear, starting from this point – not all the patients included in the constituent trials had operations, and the eligibility criteria didn't specify that patients had to be surgical. The descriptor perioperative is used in the statement of objectives at the end of the Introduction as well.

Response: *We would like to thank the reviewer for taking the time in reviewing this manuscript. All patients in the included studies had a compromised physiology; intervention, surgery and critically illness (including previous surgery), which impacts comfort of the patient and thus sleep. Furthermore these are all in-hospital patients. Therefore we found it acceptable to combine these patients. Table 2 describes the timing, duration (per session and in total) and frequency of the intervention applied. We agree that the word "perioperative" should be removed from the objectives in the abstract and did so in the revised version of the manuscript.*

We added the following section to the manuscript in the Methods (Study screening and selection section):

"Other eligibility criteria were; full text available and the study included in-hospital patients. Since the above mentioned studies included patients who have the following in common; in-hospital patients and their compromised physiology (e.g. by performing an intervention, surgical procedure, or having critical illness), which have impact on the patients comfort and thus sleep, the data was suitable to be pooled. Studies involving live music...."

1b Reading the Introduction, I was thinking that the intervention would be happening in the evening or maybe even while participants were asleep. This point wasn't clear in table 2. I think it was described in the text page 6, lines 10-30, for most of the trials. The information would be easier to assimilate if the descriptions in the second column of table 2 were expanded to include time of day, length of therapy on each occasion, number of occasions, and timing in relation to operation where appropriate. The issue of timing should be touched upon in the Introduction – that is whether its at night/evening or could be during the day– and it should be clear that the literature search put no restrictions on this. I think its only in the Discussion that they describe what constitutes poor sleep – though it's mentioned in the Introduction there is no description. I was thinking about people sleeping during the day, presumably the music therapy should be given around bedtime to reduce daytime sleeping.

Response: *We made adjustments in table 2 to make the timing of the music intervention during the day more clear by adding additional text to the “Timing intervention” column. The following parameters mentioned by the reviewers; “length of therapy on each occasion, number of occasions, and timing in relation to operation” are described in table 2 respectively under the columns; frequency (per day) x duration (min.) and timing intervention. A disturbance of the normal day-night-rhythm is a common phenomenon in patients after surgical interventions and/or on an ICU. However, this is unwanted, and it is state-of-the-art critical care to maintain the physiological day-night-rhythm as good as possible. This was the goal of all the studies included in this review. We did not mention the timing in the introduction since we wanted to investigate in general was the effect of music is on sleep regardless of the timing.*

2 Abstract, line 32/33, qualitative and quantitative analysis. In the context of a systematic review this might be taken to refer to narrative review and quantitative meta-analysis respectively. Whereas I think they are referring to analysis of the quality and quantity of sleep. This should be rephrased.

Response: *qualitative analysis refers to the risk of bias and quantitative analysis refers to the meta-analysis, as stated by the Cochrane handbook for systematic reviews. We specified this in the abstract as follows:*

“Five studies (259 patients) were included in qualitative (risk of bias) and quantitative analysis (meta-analysis).”

3 There was also a lack of clarity about the patient group the review is meant to relate to. On page 6, lines 24- 26, they mention restricting the literature search to 1981 onwards, the year when minimally invasive surgery was introduced. They don't mention anywhere that surgical patients should have had minimally invasive surgery, and it doesn't apply to non-surgical patients. Does this mean that music therapy isn't appropriate for patients having more invasive surgery? or is it that it would be difficult to carry out a trial in this wider group?

Response: *The year minimally invasive surgery was introduced, the perioperative pain protocol, which influences sleep quality, also changed. Since we specified that this review will be carried out in critically ill and surgical patients, it was important to specify this. This does not mean that patients with more invasive surgery are excluded from this review. We further specified this in the Methods under subheading "Search strategy" as follows:*

"A systematic search was performed together with a dedicated biomedical information specialist in the Embase, Medline Ovid, Cochrane Central, Web of Science, and Google Scholar databases using a standardized protocol, including articles between the 1st of January 1981 (the year in which the first minimally invasive surgical intervention was performed, in order to avoid old literature which is not compatible with the current standard in perioperative pain practice) and 27th of January 2020."

4a Page 4, search strategy section. They should broadly list the various items in the strategy in the main text, in addition to giving the source specific code as supplementary material.

Response: *based on this comment, we have adapted as follows:*

"The search strategy included terminologies related to: sleep quality/insomnia, including sleep architecture (e.g. rapid eye movement [REM] sleep) and music. The full search strategy per database is given in supplementary file 2."

4b In particular they should detail the strategy in relation to language. Looking at the supplementary material it appears the search was limited to English language papers. But on page 9, lines 41-45, they mention that Cheraghi et al was written in Persian (but one of the authors could read it). In which case the search wasn't limited to English language papers.

Response: *Indeed non-English title and abstracts were excluded, and for the Persian study there was an English abstract available and therefore we included it since full text non-English was not an exclusion criterion of our study. And since we have a statistician in the Erasmus MC with Persian as a mother language, we consulted with him and he fully translated the paper. We made the following adjustment in the Results section under subheading “Study characteristics” to make this more clear. There were no other English abstracts, with appending full text, non-English manuscripts. Therefore, there was no bias for the article selection.*

“Since the abstract of the paper of Cheraghi et al.⁶ was available in English, the study was included after title/abstract screening. The full text version was only available in the Persian language, the paper was translated by a statistician in the Erasmus MC with Persian as mother language. The study was included in this review due to acceptable methodology, which was not different than the other included studies.”

4c This is especially an issue as from Figure 1 Cheraghi et al appears to be one of the worst with respect to bias and from Figure 2 it stands out from the others with respect to the effect size reported and may be responsible for the heterogeneity found. Had this trial been excluded the overall SMD would have fallen from 1.21 to 0.87.

Response: *We would like to refer to the responses to the comment nr. 2 of reviewer 3 as answer to this question, where we dealt with this point by means of a sensitivity analysis excluding this study.*

5 Page 6, lines 41-48. Please could you explain what PICOS stands for.

Response: *An explanation with reference is provided in the methods under subheading “study screening and selection” as follows:*

“Studies were eligible when they investigated (P) critically ill and/or surgical patients aged 18 years or older receiving a (I) recorded music intervention (C) compared to a control group in order to assess the effect on (O) sleep quality and quantity in (S) a randomized clinical trial (RCT). PICOS is a mnemonic used in evidence based medicine and stands for, respectively; Patient, Intervention, Control, Outcomes, and Study type.”

6 Page 6, line 53/4. Studies involving live music were excluded. Please could you give a justification as to why such trials don't match the objectives of the current study.

Response: *Live music performance consists of two interventions; the music and the interaction with the musician, this does not depict the pure effect of music on sleep. We adapted as follows in Results (Study screening and selection):*

“Studies involving live music performance were excluded, since live music performance consists of two interventions; the music and the interaction with the musician.”

7a Page 7, line 22-24. Abstracting information on population size. From Table 1, heading to column 5, its clear that by population size they mean the size of the trial. Usually population refers to a wider group from which people are selected to participate in the study. So it would be clearer on page 7 (and in table 2) to refer to the size of the trial.

Response: *We agree that “Population” is unclear, thus this is replaced with “Sample” throughout the manuscript and in Table 1.*

7b When they describe the eligibility criteria it's possible to include a trial with more than two groups – possibly a third arm could have a different intervention to improve sleeping – and in such a case only the arms closest to music therapy and control would be included in the meta analysis. I notice from Tables 1 and 2 that the “population size” is equal to the total of participants in the music therapy and control arms in all cases. Does that mean that none of these trials had additional arms that weren't included, or does it mean that “population size” means the total of the music therapy and control arms, and not the size of the trial (in groups with more than two arms)?

Response: *“population size” refers to the total of the music and control arms.. We added the following in the Results (Study screening and selection) to specify that we did not use the data of the extra arms of those (only Zimmerman et al) this:*

“If studies compared music to multiple groups, the group without music most similar to the music group was chosen as control group (e.g. if groups were ‘scheduled rest’ and ‘standard care’, ‘scheduled rest’ was chosen as control group if the music group also received a resting period). The data of the extra arms were therefore not used in this study. Full text articles were discussed for admissibility. All disagreements between reviewers were resolved by discussion.”

8 Page 7, lines 37-41, the selection of data relating to the final time point to include in the study, because music therapy could have a cumulative effect. The number of participants still in trial follow-up tends to decrease with successive follow-ups. They do say somewhere that attrition in the trials was low (less than 10%), but perhaps they would have had analysis based on a larger number if they had selected an earlier timepoint. Another point is that though the impact of therapy might be expected to be cumulative with therapy sessions, it might also be expected to tail off after therapy has finished, and it might have been preferable to have planned to take the follow-up closest to the end of therapy. The timing of the assessment used, is described on page 10, lines 10-30 (the section that also described the timing of therapy). It would be easier to understand the timing of the assessment used if this was included in Table 1 – perhaps in the outcome assessment column. The way the timing of assessment was described on lines 10-30, didn't sound as though there were multiple timepoint and the final one was selected as stated on page 7, lines 37-41.

Response: *We thank the Reviewer for the suggestions made. We added a paragraph in the Discussion (Future research recommendations), because adhering to these ideas will reduce heterogeneity.*

“Future research recommendations

We suggest future research should consist of high quality randomized clinical trials with the use of objective tools for sleep assessment, as recorded music seems effective and clinically relevant, in order to make more definite conclusions regarding the effect of recorded music on sleep. Since, until now relatively small studies with a high variability in the music “dose” are conducted we recommend future studies to focus on larger sample sizes with a high methodological quality in order to avoid a substantial risk of bias. Studies with music interventions should report the type of music, timing, duration, and frequency of the intervention and sleep assessment in their studies with a validated and reproducible tool. Our analyses suggest that a minimum of 30 minutes per day/session is sufficient in order for the music intervention to be effective for the sleep quality. This minimum of 30 minutes of music per session/day is also consistent with the current literature. Previous studies can serve as a guide for future studies.”

Regarding the timing of assessment, this is added to table 2 and we also further specified this in the Results;

“Cheraghi et al. administered the music intervention during three consecutive evenings, just before bedtime, after admission to the CCU, and assessed sleep at baseline and every morning after an evening of intervention. In the study of Su et al. the intervention was only administered in the night of day three after admission to the ICU and sleep was assessed at baseline (PSG and VSH) and PSG during the first 2 hours of sleep on the same night and VSH in the morning after the night of intervention. Hansen et al. applied the intervention during the day on day three after admission and assessed sleep once immediately afterwards. Zimmerman et al.⁷ applied the intervention on the night of postoperative day (POD) two and assessed sleep at baseline and the morning of POD three.”

9a Page 8, lines 14-18, how mean age was obtained where age was given in categories. They say the mean for each category was calculated and pooled. But if only numbers in each category are available category means can't be calculated. They could have taken the central point of the age range in each category and averaged these, weighting by category frequencies?

Response: *We thank the Reviewer for this comment; we calculated as follows: 20-50 = 8 patients, 51-60 = 7. The mean age would be calculated as followed: the central point of 20-50 is 35 and the central point of 51-60 is 55.5. The mean age would be $((35*8)+(55.5*7))/(8+7) = 44.6$. We rephrased this in order to make it more clear in Methods:*

“For categorical age groups (e.g. age 20-50 = 8 patients, 51-60 = 7 patients) an approximation of the mean was calculated by taking the central point of the range as the mean for each category and pooling these means for each trial separately, weighted to the sample size.”

9b I assumed this related to the summary ages and genders at the bottom of Table 1. Did they obtained the trial specific mean ages then average over the 5 trials – and was this averaging weighted according to the size of the trial?

Response: *The Reviewer is right; we did so.*

9c They could also mention how the overall % male were obtained – this could also have been obtained by averaging trial specific %s.

Response: *We thank the reviewer for this suggestion and added the specification to this In Methods as follows; “The overall % of males in this review was calculated by averaging the % of males weighted to the sample size.”*

10 Page 8, line 22, quantitative analysis. Its not clear whether they mean the analysis of the quantity of sleep here, or whether quantitative analysis is being used to mean meta analysis itself (as in my point 2).

Response: *We feel, that there might be a misunderstanding. According to the PRISMA statement, a qualitative meta-analysis covers non-calculatable findings and a quantitative analysis covers the outcomes which can be calculated. The quality and the quantity of sleep, both, can be calculated. We would like to refer to the response to the previous comment 2 of this Reviewer.*

11 Page 8, line 28/29, the choice to perform random effects meta analysis. This is most sensible when there are a large number of studies to be combined. Here they have only 5 (4 if Chereghi et al is excluded). A small number of studies is often taken as a reason to do fixed effects meta analysis. Did a fixed effects analysis result in different conclusions?

Response: *We thank the reviewer for this comment. We assume the reviewer was referring to a fixed effect model, which is based on a common effect across all studies with 0% heterogeneity, and not a fixed effects model in which a separate effect is estimated for each study –the latter method is seldom used in meta-analysis. Although the number of studies (4 or 5) in our meta-analysis leads to considerable uncertainty in the estimated heterogeneity in the random effects model, this does not mean that it is a good idea to assume that the heterogeneity is 0 (as in the fixed effect model). In fact, using random effects models is quite standard in meta-analysis, even if the number of studies is lower than 4. We did perform the fixed effect meta-analysis, which gave the following results; 5 studies: SMD 1.16 [95% CI 0.89; 1.43] $p < 0.0001$; 4 studies (excluding Cheraghi): SMD 0.85 [95% CI 0.54; 1.15] $p < 0.0001$. Because we prefer the random effects model, these results are however not reported in the manuscript.*

12 Page 8, line 24/5 onwards the main method of meta analysis. I would imagine that many of the constituent trials reported baseline values of the instruments chosen as their outcome measure, taken

at the point of randomisation before the start of intervention. Was anything done with such information? IF not they should state that this information was not used.

Response: *We certainly looked at baseline values. Only the study of Cheraghi, Su and Zimmerman reported baseline values for sleep assessment, therefore only the post intervention values were used for meta-analysis. We added a sentence in the Results to specify this.*

“Baseline sleep assessment was carried out in only three studies, no differences were found between the study groups in any studies based on baseline data.”

13 Page 8, line 30/31. The “in” as in “the in between study variance was estimated...” can be omitted. The between study variance won’t be well estimated with only a few studies.

Response: *We removed the “in” in this sentence.*

14 Page 8, lines 37-47, the description of clinical interpretation. I didn’t understand exactly what was done here. The description of the method continued on page 11, lines 4-7 in the Results section. I thought a standardised mean difference could be interpreted easily as the size of the difference in terms of SD between the groups. The methods described lead to their estimate of a 27.1% increase in sleep quality, the focus of conclusions (eg page 14, line 9/10) and in the Abstract line 39 where it is stated as a 26% increase. I felt the methods described were quite involved and based on assumptions – couldn’t they concentrate on the SMID finding?

Response: *We would like to refer to the response to comment 2 of Reviewer 2. We thank the Reviewer for the scrutiny and apologize for the error. We adjusted the 26% to 27.1%.*

15 Page 9, lines 50-56, the description of the instruments used to measure outcome in the trials. It would be helpful if some description of these tools were given. In the Discussion there is a description of the various aspects of poor sleep quality. I wanted to know whether these instruments (and the patient questionnaire used by Ryu et al) measured the same underlying constructs. In particular I note from Table 1 that Chereghi et al was the only one using the PSQI, if this assesses different aspects to the others it might explain the extreme result from this trial. If the instruments don’t measure the same thing they shouldn’t be combined, the authors should address whether they are similar enough to combine.

Response: *we added Supplementary file 3 which gives an overview of the sleep questionnaire characteristics used in the studies of our review and the following section to the Results (Study characteristics):*

“Sleep quality was measured using the Richards-Campbell Sleep Questionnaire (RCSQ, 40%), the Verran and Snyder-Halpern sleep scale (VSH, 40%), and the Pittsburgh Sleep Quality Index (PSQI, 20%). Supplementary file 3 gives an overview of the previously mentioned sleep questionnaire characteristics. Only in the study of Ryu et al. sleep quantity was measured using patient questionnaires. In the study of Su et al. sleep quality and quantity was also measured objectively, using the polysomnography (PSG). PSG is currently the golden standard in objectively measuring sleep variables, including total sleep time (TST), wake after sleep onset (WASO), sleep onset latency (SOL), REM latency, sleep efficiency (SE), arousal index (AI), and percentage of total sleep time spent in each sleep stage (N1, N2, N3, and REM).”

All these questionnaires assess sleep health, which in the studies added to the meta-analysis is called sleep quality. Also, these questionnaires are widely used to assess sleep quality in hospitalized patients, see reference 86. We also added a section in the discussion (strengths and limitations) regarding this:

“All of the three tools used in the studies included in the meta-analysis are validated and reliable self-reporting questionnaires assessing sleep health of the previous night, and are used to assess sleep quality in the studies included in this meta-analysis. The tools are widely used to measure sleep quality in hospitalized patients, and thus are deemed conceptually similar enough to perform meta-analysis.”

16 Page 10, line 22. I don't think the acronym POD has been spelt out in the text. I notice several acronyms are spelt out in footnotes to the tables, but I think they should be spelt out in the text as well.

Response: *Thank you for this scrutiny: we checked all the acronyms in the manuscript and spelt these out where it was not done yet; ACS, SD.*

17 Page 10, lines 45-50. The analysis excluding the Chereghi trial. It would be useful at this point to comment on the heterogeneity results (i squared and t squared). In particular they could report corresponding results from the analysis excluding Cherggi et al. Was there an indication of heterogeneity after the trial was removed? Given that this trial did not meet the stated eligibility criteria (it was not published in English), it might be better to emphasise the results excluding this trial in the Abstract and conclusions.

Response: *Thank you for your suggestion. We added the forest plot without the study of Cheraghi as Figure 2B to the manuscript. We also made changes in the abstract and Results of the main text in order to emphasize this.*

18 Page 11, lines 5-10, Chereghi et al excluding participants where the therapy induced anxiety etc. Could they state the numbers of participants that were excluded for this reason. IN connection to my point 7 above relating to the population size variable, the size of the trial. Do the numbers given in Tables 1 and 2 relate to the number of participants randomised in the constituent trials, or the numbers included in this meta analysis? It would be possible to include both in the tables.

Response: *We would like to refer to the answer of point 7 of this reviewer.*

19 Table 1. Please could they include the reference number for each trial after the study names in column 1.

Response: *we added the references to the table.*

VERSION 2 – REVIEW

REVIEWER	michele umbrello AO San Paolo - Polo Universitario
REVIEW RETURNED	07-Feb-2021

GENERAL COMMENTS	The authors adequately responded to my comments. The only concern is that it seems that the order of the authors has been changed in the current revised version (Dr J. Jeekel, and Dr. M. van der Jagt switched their place): do all the authors agree with this change? What is the reason for such change?
---

REVIEWER	Michele Balas Ohio State University College of Nursing, USA
REVIEW RETURNED	24-Jan-2021

GENERAL COMMENTS	Appreciate the time and effort the authors took responding to the comments in their response to the last review. I believe the article is now acceptable for publication. My only minor concern relates to line 63-64 (seems odd and rather
---

	irrelevant. Would recommend deleting
REVIEWER	Irina Chis Ster St George's University of London
REVIEW RETURNED	08-Feb-2021
GENERAL COMMENTS	My queries were satisfactory addressed - hence I am happy to recommend publication.